# A dual-interface load transfer model for precast concrete-cored cemented soil piles considering progressive damage

Haian Liang[1,2], Yating Zhu[1]*, Lanlan Xu[1], Hongyan Wu[1], Junpeng Zhang[1], Bin Xu[3], Kaiwei Cao[4]

1 School of Civil and Architectural Engineering, East China University of Technology, Nanchang, Jiangxi, China, 2 Jiangxi Provincial Research Center for Geological Environment and Underground Space Engineering, Nanchang, Jiangxi, China, 3 School of Civil and Architectural Engineering, Jiangxi University of Water Resources and Electric Power, Nanchang, Jiangxi, China, 4 Jiangxi Zhongheng Underground Space Technology Co., Ltd., Nanchang, Jiangxi, China

* 2602729736@qq.com

## Abstract

Precast concrete-cored cemented soil piles (PCCS) are widely used to improve the bearing capacity of soft ground. However, existing design models usually assume perfect bonding between the concrete core and cemented soil shell, ignoring progressive interface damage under external loads, which leads to inaccuracies in bearing capacity predictions. This study proposes a dual-interface load transfer model that explicitly simulates progressive interface damage. Its innovation lies in the coupling of an exponential damage constitutive model for the inner concrete–cemented soil interface calibrated by direct shear tests with an elastoplastic model for the outer cemented soil–surrounding soil interface, along with a convergent iterative algorithm for solving the governing equations. Verified by field measurements of the test pile in Jiangxi Province, China, the model shows strong agreement with measured data $R^2 = 0.9746$. Quantitative analysis shows that the concrete core bears more than 90% of the pile-head load, and the cemented soil shows a distinctive C-shaped axial force distribution along the pile shaft. The model captures the coupled evolution of load sharing and skin friction at both interfaces, providing theoretical support for the refined design and safety assessment of PCCS foundations.

## 1 Introduction

Precast concrete-cored cemented soil piles (PCCS), consisting of a high-stiffness precast concrete (PC) core encased in a cemented soil shell, have been widely adopted in geotechnical engineering for soft ground improvement, high-rise building foundations, and highway embankment reinforcement [1–4]. This composite configuration leverages the high axial stiffness of the concrete core and the favorable interfacial bonding properties of the cemented soil, thereby enhancing bearing capacity

**Data availability statement:** All relevant data are within the paper and its Supporting Information files.

**Funding:** This work was supported by the National Natural Science Foundation of China (Grant No. 52168045) and the Key Research and Development Program of Jiangxi Province(Grant No. 20252BCF320006). There was no additional external funding receivedfor this study. The funders had no role in study design, data collection and analysis, decision to publish, or preparation of the manuscript.

**Competing interests:** The authors have declared that no competing interests exist.

and controlling settlement more effectively than conventional single-material piles in complex geological conditions [5–6]. The configuration and construction process of the PCCS are shown in Fig 1.

Nevertheless, the design and construction of PCCS involve complicated pile-soil interactions. Their vertical bearing behavior is fundamentally governed by load transfer across two interfaces: the PC pile–cemented soil interface and the cemented soil–surrounding soil interface. However, most existing models fail to capture the coupled evolution of damage at these interfaces. Accurately characterizing these coupled interfacial behaviors remains a critical challenge, as neglecting interface damage may result in overestimation of bearing capacity (compromising structural safety) or excessively conservative designs (reducing cost-effectiveness). Therefore, a model that explicitly accounts for the evolution of interface damage is essential for both structural safety and economic efficiency.

Over the past two decades, scholars worldwide have investigated the load transfer characteristics of PCCS using theoretical, experimental, numerical and analytical methods [7–12]. Based on classic load transfer theories [13–14], some researchers assumed perfect bonding between the core pile and cemented soil and only considered shear failure at the pile-soil interface, establishing simplified single-interface models [15–21] that are inconsistent with the actual mechanical behavior of PCCS. Although existing studies have adopted advanced interface constitutive models featuring nonlinear softening and modulus degradation [22–23], their application is limited to single-interface systems and has not been extended to PCCS with dual-interface characteristics. Meanwhile, studies that consider the dual-interface effect still simplify the interface constitutive relationship to linear elastic or perfectly plastic models, which cannot reflect the progressive degradation of the interface under loading. Subsequent investigations have further improved the understanding of the load transfer mechanism of PCCS through field tests, numerical simulations and improved calculation methods [24–29]. Despite these contributions, significant limitations still exist. Most existing theoretical models assume either a single interface or perfect bonding between components, neglecting the coupled interaction between the two interfaces as well as the residual shear resistance after interfacial slip or damage. Such simplifications compromise the accuracy and generalizability of analytical predictions, limiting their application in refined engineering design.

To overcome these limitations, this study proposes a load transfer model for PCCS that explicitly considers the coupled behavior of both interfaces. The proposed model integrates an ideal elastoplastic representation of the cemented soil–surrounding soil interface with a nonlinear damage constitutive law calibrated from direct shear tests for the PC pile–cemented soil interface. This dual-interface framework is formulated as a set of governing equations that describe the entire load transfer process in the PC pile–cemented soil–surrounding soil system. An iterative calculation method with convergence characteristics is established to solve the mechanical response under axial loading. The model is validated against field measurements from an instrumented test pile, demonstrating its reliability and practical value for the design and performance evaluation of PCCS.

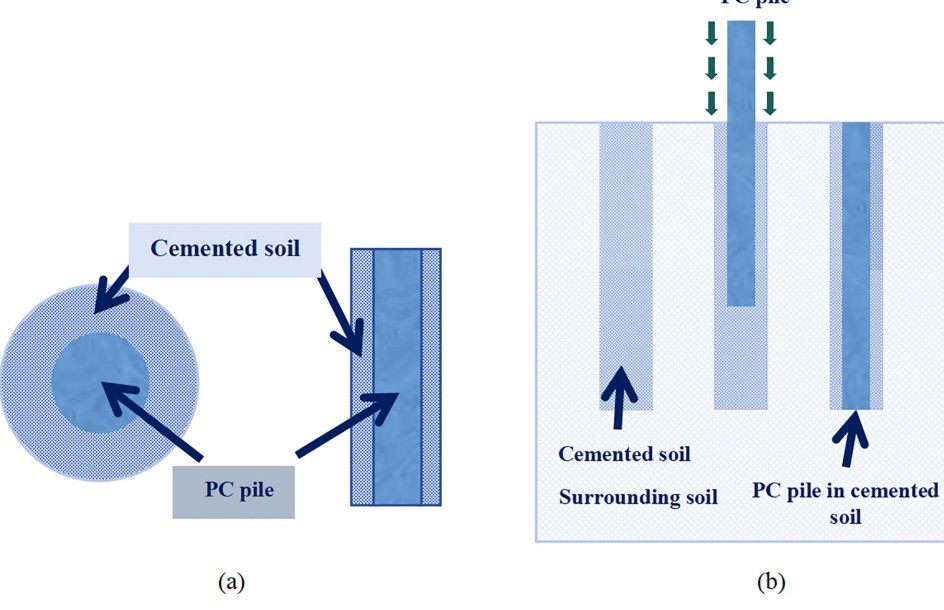

**Fig 1. Configuration and construction process of the PCCS (a) Configuration of the PCCS (b) Construction process of the PCCS.**

## 2 Mechanical model and basic assumptions

### 2.1 Simplified dual-interface mechanical model

Fig 2 shows the simplified dual-interface mechanical model for load transfer of PCCS. Under vertical loading, the external load is transferred and redistributed through interactions among the PC pile, cemented soil, surrounding soil, and their two interfaces [30]. Due to its higher stiffness, the PC pile initially bears most of the external load. The resulting settlement

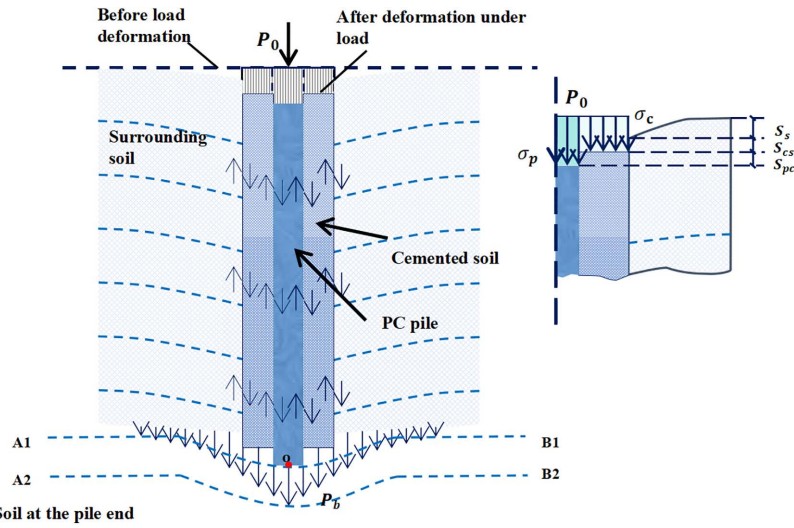

**Fig 2. Schematic of the dual-interface coupling model for PCCS.**

induces relative displacement at the PC pile–cemented soil interface, thereby mobilizing shear stress. As the load increases, this interface undergoes progressive stiffness degradation and damage accumulation.

Subsequently, the load is transmitted into the cemented soil and further transferred to the surrounding soil through the cemented soil–surrounding soil interface, where the response gradually transitions from elastic behavior to plastic slip. As the load propagates downward along the pile shaft, the mobilized skin friction at both interfaces resists most of the axial load, with only a small fraction reaching the pile tip.

Because the PC pile settles faster than the cemented soil, shear stress $\tau_p$ is mobilized at their interface. Similarly, the cemented soil settles faster than the surrounding soil, mobilizing shear stress $\tau_c$ at the cemented soil–surrounding soil interface. The total pile-head settlement $S(z)$ is the sum of the relative displacements at both interfaces and the shear deformation of the surrounding soil [31]:

$$S\left(z\right) = s_{pc} + s_{cs} + S_s$$

(1)

Where $S\left(z\right)$ denotes the total pile-head settlement; $s_{pc}$ denotes the relative displacement at the PC pile–cemented soil interface; $s_{cs}$ denotes the relative displacement at the cemented soil–surrounding soil interface; $S_s$ denotes the shear deformation of the surrounding soil.

## 2.2 Basic assumptions

Based on the above analysis, the following assumptions are adopted for model development:

1. The PC pile, cemented soil, and surrounding soil are treated as homogeneous, isotropic, and linearly elastic materials (Hooke's law applies prior to yielding or damage).

2. Load transfer at the PC pile–cemented soil interface follows the exponential damage model, whereas that at the cemented soil–surrounding soil interface conforms to the ideal elastoplastic model.

3. The pile tip is idealized as a rigid foundation on an elastic half-space, with the surrounding soil undergoing axisymmetric radial shear deformation.

Based on the above assumptions, the PC pile, cemented soil, and surrounding soil are regarded as linearly elastic materials, which is a common simplification in analytical models for composite piles. This idealization facilitates theoretical derivation and calculation efficiency, yet it ignores the heterogeneity, anisotropy, and nonlinearity of natural soils. Therefore, the proposed model is appropriate for preliminary design and engineering estimation under small deformation conditions.

## 2.3 Interface constitutive models

Experimental investigations on the shear behavior of PCCS interfaces [32] indicate that the exponential damage model captures two key mechanical features: (1) the nonlinear damage accumulation process from microcrack initiation to complete failure, and (2) the relationship between interfacial shear stress $\tau_p$ and relative displacement $s_{pc}$ at the PC pile–cemented soil interface. Accordingly, the behavior of the PC pile–cemented soil interface is described using an exponential damage model, as illustrated in Fig 3(a). The corresponding shear stress-displacement relationship is expressed as:

$$\tau_p = \begin{cases} \tau_{u1}\left(1-e^{\frac{-k_1 s_{pc}}{\tau_{u1}}}\right) & s_{pc} \leq s_{u1} \\ \tau_{u1}e^{-(s_{pc}-s_{u1})^m}+\tau_r & s_{pc} > s_{u1} \end{cases}$$

(2)

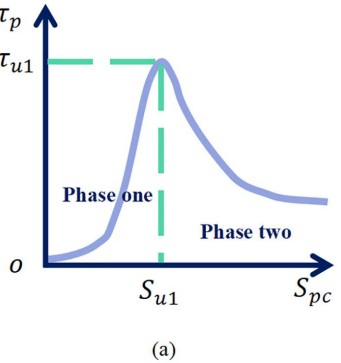 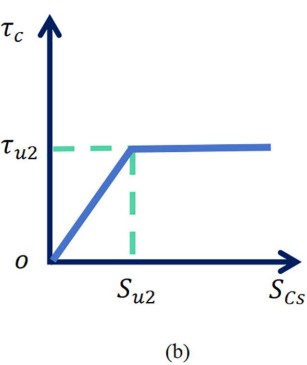

(a)                                                    (b)

**Fig 3. Dual-interface system behavior: shear stress versus displacement (a) PC pile-cemented soil interface (b) Cemented soil-surrounding soil interface.**

where $\tau_p$ and $s_{pc}$ denote the interfacial shear stress and relative displacement at the PC pile–cemented soil interface, respectively; $\tau_r$ is the residual shear strength; and m is the Weibull modulus. Based on statistical damage theory, the model assumes that the strength of the micro-elements at the interface follows a Weibull distribution, where m characterizes the material brittleness. The parameter $k_1$, representing the interfacial stiffness coefficient, is positively correlated with the normal stress acting on the contact surface.

Previous studies [33–34] have validated that the elastoplastic model effectively captures the load-transfer mechanism at interfaces between cement-based materials (e.g., cemented soil) and the surrounding soil. Accordingly, the cemented soil–surrounding soil interface is idealized as an elastoplastic model, as shown in Fig 3(b). The corresponding shear stress-displacement relationship is given by:

$$\tau_c = \begin{cases} k_2 s_{cs} & s_{cs} \leq s_{u2} \\ \tau_{u2} & s_{cs} > s_{u2} \end{cases}$$

(3)

where $\tau_c$ and $s_{cs}$ are the interfacial shear stress and relative displacement at the cemented soil–surrounding soil interface, respectively; $\tau_{u2}$ and $s_{u2}$ denote the ultimate interfacial shear stress and the corresponding critical relative displacement; and $k_2$ represents the interfacial shear stiffness.

Within a certain range along the pile side, the shear deformation of the surrounding soil is given by: $s_s(z) = \frac{\tau_c(z) r_c \ln(\frac{r_m}{r_c})}{G_s}$, where $r_c$ is the radius of the cement-soil pile, $G_s$ is the shear modulus of the soil on the pile side, and $r_m$ is the radius of influence, taken as $r_m = 2.5 \, \rho l (1 - v)$, with ρ being the soil heterogeneity coefficient, which is taken as 1 when the soil is homogeneous.

## 3 Load transfer equations and calculations

### 3.1 Calculation element analysis

An infinitesimal linear-elastic element at depth z was extracted from the system, encompassing the PC pile, cemented soil, and surrounding soil. The force-deformation responses of the two internal interfaces (the PC pile–cemented soil interface and the cemented soil–surrounding soil interface) were analyzed separately, as shown in Fig 4.

The static equilibrium condition of the linear-elastic element yields:

$$\begin{cases} d\sigma_p(z) = \frac{-\tau_p(z) u_p}{A_p} dz \\ d\sigma_c(z) = \frac{-\tau_c(z) u_c}{A_c} dz + \frac{\tau_p(z) u_p}{A_p} dz \end{cases}$$

(4)

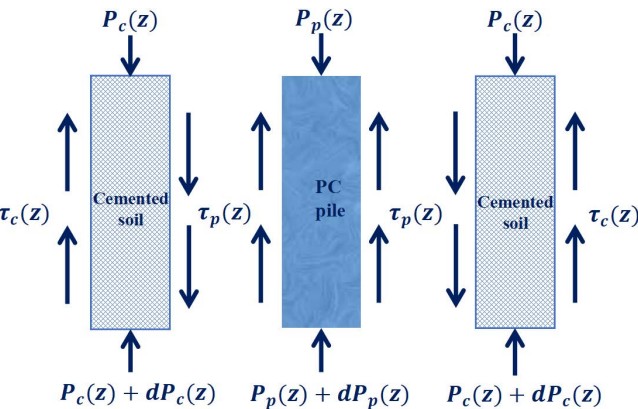

**Fig 4. Elastic element force analysis.**

where $\sigma_p(z)$ and $\sigma_c(z)$ are the axial stresses of the PC pile and cemented soil, respectively; $\tau_p(z)$ and $\tau_c(z)$ are the interfacial shear stresses; $u_p$ and $u_c$ are the perimeters; and $A_p$ and $A_c$ are the cross-sectional areas of the PC pile and cemented soil, respectively.

Given the elastic compressive deformations $\Delta_p$ and $\Delta_c$ of the PC pile and cemented soil elements, the corresponding relative displacement increments are given by $ds_{cs}(z) = \Delta_c$ and $ds_{pc}(z) = \Delta_p - \Delta_c$, as shown in Fig 5.

The compressive deformations at the two interfaces (between the PC pile and cemented soil, and between the cemented soil and surrounding soil) are derived as follows:

$$\begin{cases} ds_{pc} = \frac{\sigma_p}{E_p}dz - \frac{\sigma_c}{E_c}dz \\ ds_{cs} = \frac{\sigma_c}{E_c}dz \end{cases}$$

(5)

where $E_p$ and $E_c$ are the elastic moduli of the PC pile and cemented soil, respectively.

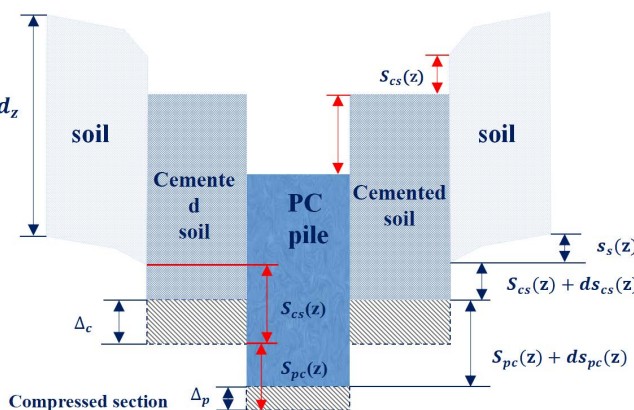

**Fig 5. Schematic diagram of stress and deformation of the elastic element.**

## 3.2 Establishment of load transfer relationships

The pile is divided into m elements of equal length $\Delta L = \frac{L}{m}$, generating m + 1 nodes. Here, Node 1 is located at the pile toe ($z = L$) and Node m + 1 at the pile head ($z = 0$). The recursion is implemented upward from the pile toe, since the displacement at the pile toe is generally the smallest along the pile shaft. In this model, the analysis is initiated by prescribing the pile toe displacement as a boundary condition. Carrying out the recursive calculation from the pile toe upward can minimize errors introduced by the assumed displacement boundary and thus better represent the actual mechanical response of the pile.

1. For any node i satisfying $1 < i \leq m+1$, the axial stress increments $d\sigma_p(z)$ and $d\sigma_c(z)$ in the PC pile and cemented soil elements between nodes i and i + 1 are given by:

$$\begin{cases} d\sigma_p(z) = \sigma_{p(i)} - \sigma_{p(i+1)} \\ d\sigma_c(z) = \sigma_{c(i)} - \sigma_{c(i+1)} \end{cases}$$
(6)

where $\sigma_{pi}(z)$ and $\sigma_{ci}(z)$ are the axial stresses in the PC pile and cemented soil at the i-th node, respectively.

Substituting Eq. (5) into Eq. (6) yields the stress values at node i + 1 as follows:

$$\begin{cases} \sigma_{p(i+1)} = \sigma_{p(i)} + \frac{\tau_{p(i)} u_p}{A_p} \Delta L \\ \sigma_{c(i+1)} = \sigma_{c(i)} + \frac{\tau_{c(i)} u_c}{A_c} \Delta L - \frac{\tau_{p(i)} u_p}{A_p} \Delta L \end{cases}$$
(7)

Based on the mechanical compatibility of relative displacements at the PC pile–cemented soil and cemented soil–surrounding soil interfaces, the following expressions are derived:

$$\begin{cases} s_{p(i+1)} = s_{p(i)} - \frac{\sigma_{c(i)}}{E_c} \Delta L + \frac{\sigma_{p(i)}}{E_p} \Delta L \\ s_{c(i+1)} = s_{c(i)} + \frac{\sigma_{c(i)}}{E_c} \Delta L \end{cases}$$
(8)

2. Following Randolph et al. [35], the pile toe is modeled as a rigid punch resting on an elastic half-space. Its boundary conditions satisfy:

$$\begin{cases} P_p(L) = k_L A_p s_p(L) \\ P_c(L) = k_L A_c s_c(L) \end{cases}$$
(9)

where $P_p(L)$ and $P_c(L)$ denote the base resistances of the PC pile and cemented soil, respectively. The stiffness coefficient of the underlying soil at the pile toe is given by $k_L = \frac{4G_L}{[\pi r_c(1-v_L)]}$, where $G_L$ is the shear modulus and $v_L$ is the Poisson's ratio of the soil at the pile toe level.

By assuming the displacement and deformation at the pile toe, the pile toe stress and skin friction are obtained according to Eqs. (9), (2), and (3). Substituting these into Eqs. (4) and (5) yields the stress increment and displacement increment between Node 1 and Node 2. Combined with Eqs. (6), (7), and (8), the relative displacement, deformation, and pile-body stress at the second node are derived. Then, through upward recursion from the pile toe, the displacement, deformation, axial stress, and skin friction at all m + 1 nodes are calculated recursively.

## 3.3 Iterative solution via displacement compatibility method

The displacement compatibility method ensures stress and displacement compatibility at each pile element when solving the load-transfer problem. This approach clearly reveals the stress state and load-transfer mechanism along the pile depth and offers considerable computational flexibility for parametric studies. The solution procedure is as follows:

 

1) First, the system composed of the PC pile, cemented soil, and surrounding soil is discretized into m elements along the pile length, with the pile toe designated as Node 1. This results in a total of m + 1 nodes up to the pile head, each with a length of $\Delta L = \frac{L}{m}$.

2) Assume the displacement value of the PC pile at the pile toe ($i = 1$) is $s_{p(i)}^{(1)}$ (the superscript denotes the iteration number), and select the displacement value of the cemented soil at the pile toe as $s_{c(i)}^{(1)}$ for the first iteration. Moreover, the inequality $0 < s_{c(i)}^{(j)} < s_{p(i)}^{(j)}$ (where j denotes the iteration number) must be satisfied throughout the iterative process.

3) The axial stress values of the PC pile and cemented soil at Node 1 are calculated separately using the boundary conditions given in Eq. (9).

4) The assumed displacement values are substituted into Eqs. (2) and (3), respectively, to obtain the dual-interface skin friction and soil shear deformation at Node 1. Subsequently, the obtained skin friction values at Node 1 are substituted into Eqs. (4) and (5) to derive the stress increment and displacement increment between Node 1 and Node 2.

5) Combining the axial stresses and displacements of the PC pile, cemented soil, and surrounding soil at Node 1 (obtained from boundary conditions) with the previously derived stress and displacement increment expressions, upward iteration based on Eqs. (6), (7), and (8) solves for the axial stresses and displacements at Node 2. Substituting the known displacement at Node 2 into Eqs. (2) and (3) yields the skin friction and surrounding soil shear deformation at Node 2.

6) This iterative propagation continues upward until the relative displacements at both interfaces and the shear deformation of the surrounding soil at the pile head (Node m + 1) are determined. The pile-head settlement $S^{(1)}$ of the PC pile is then derived.

7) At this point, the first iteration is completed, yielding the pile-head loads $P_{p(m+1)}^{(1)}$ and $P_{c(m+1)}^{(1)}$. The iteration terminates upon convergence if $| P_p^{(j)}(m+1) + P_c^{(j)}(m+1) - P_0 | < \varepsilon$, where $P_0$ is the actual pile-head load and $\varepsilon$ is the specified relative error tolerance (adopted as 1%–5% in this study). If the convergence criterion is not satisfied, the assumed pile-toe displacement is adjusted, and the next iteration is performed.

8) Repeat the above steps until the result of the j-th iteration satisfies the convergence criterion, at which point the calculation terminates.

The proposed theoretical model is suitable for both homogeneous and layered soils, with implementation achieved by adjusting the interface parameters accordingly. The iterative procedure can be implemented in MATLAB or Python. The program requires input of geometric and material parameters for the PC pile and cemented soil, the shear modulus and Poisson's ratio of the surrounding soil, and initial displacement values at the pile toe. It then automatically performs the iteration and outputs the distributions of axial stress, skin friction, displacement, and settlement along the pile depth. The program flowchart is presented in Fig 6.

## 4 Engineering case and interface parameter calibration

### 4.1 Project overview and stratigraphic conditions

The test pile is located at the factory site of Jianhua Building Materials Co., Ltd. in Nanchang County, Jiangxi Province. The inner core of the PCCS is made of C105 concrete with a diameter of 500 mm, and the outer shell is made of cemented soil with a diameter of 850 mm. The pile length is 14 m. The investigation report indicates that the soil stratigraphy is predominantly loose sand. Field site access was approved by Jianhua Building Materials Co., Ltd. No permits were required as the study did not involve endangered or protected species.

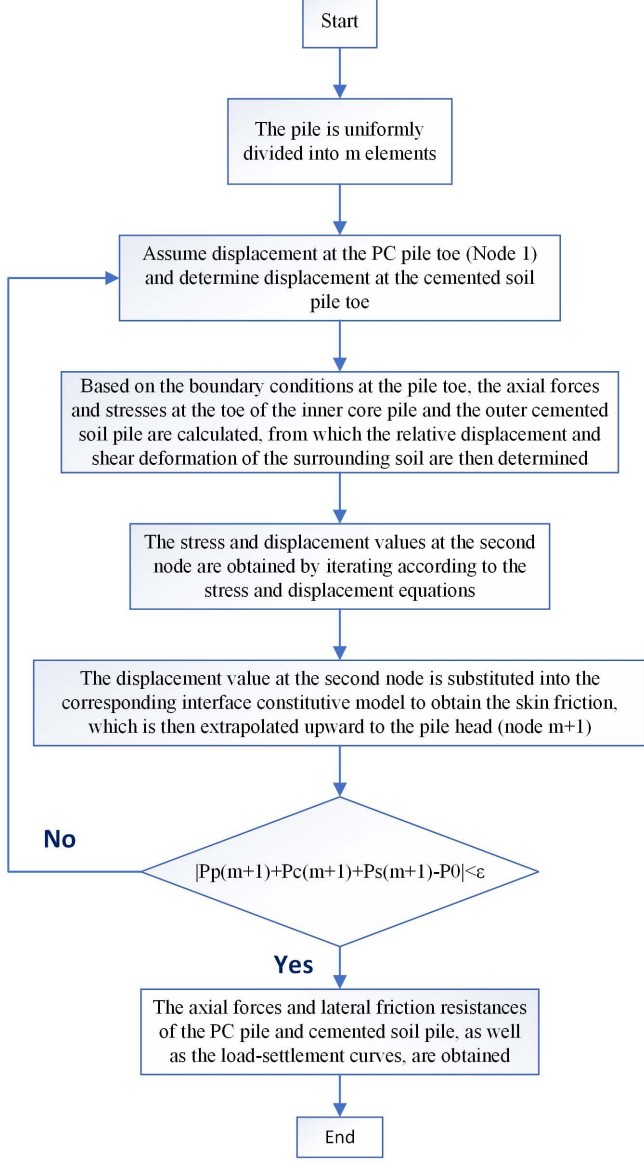

**Fig 6. Iterative calculation flowchart for displacement.**

To verify whether the exponential damage model and the ideal elastoplastic model can characterize the shear displacement curves of the dual interfaces, and to obtain the key load transfer parameters, interface shear tests were conducted.

### 4.2 Calibration of interface shear parameters

**4.2.1 PC pile–cemented soil interface shear test.** Based on an engineering case and similarity theory, a scaled-down model of the PCCS pile was designed and fabricated for laboratory tests. The geometric similarity ratio between the model and the prototype pile is 1:14, and micro-concrete is used to simulate the prototype material to ensure that the elastic modulus and compressive strength meet the similarity criteria. Strain and Poisson's ratio, as dimensionless parameters, can be directly applied to the analysis of full-scale piles. The inner core is made of C25 concrete, and three

grades of cemented soil are used as the outer shell material, with unconfined compressive strengths of 0.884, 1.974, and 3.338 MPa, respectively. The cemented soil is prepared using P·O 32.5 grade Ordinary Portland cement with a water-cement ratio of 1.0. Both the inner core concrete and the cemented soil were wet cured for 28 days. Interfacial shear specimens are fabricated by casting the concrete inner core against each grade of cemented soil to form the contact surface. The loading device is the "SHT405" microcomputer-controlled electro-hydraulic servo pressure testing machine. The prepared model pile and loading method are shown in Fig 7.

As presented in Fig 8, under constant confining pressure, increasing the cemented soil strength from 0.884 MPa to 3.338 MPa enhances the peak interfacial shear stress by 277.35% compared to the baseline value of 0.412 MPa [36].

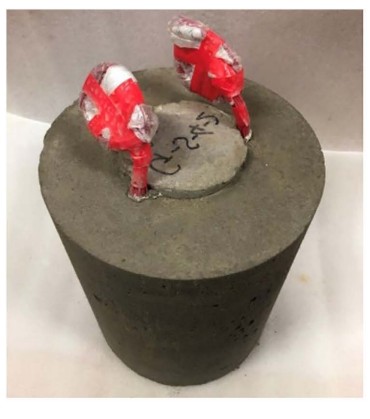 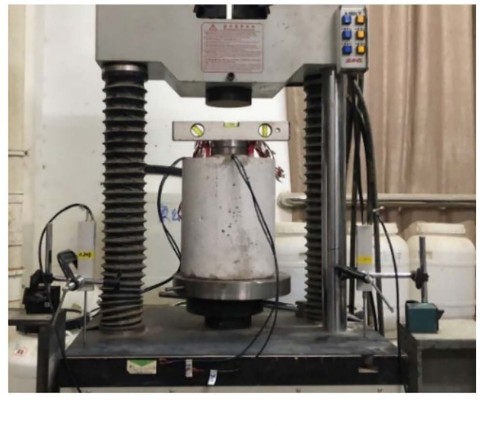

(a) (b)

**Fig 7. Specimen and shear device (a) Cement-solidified sand (b) "SHT405" micro-controlled electro-hydraulic servo pressure tester.**

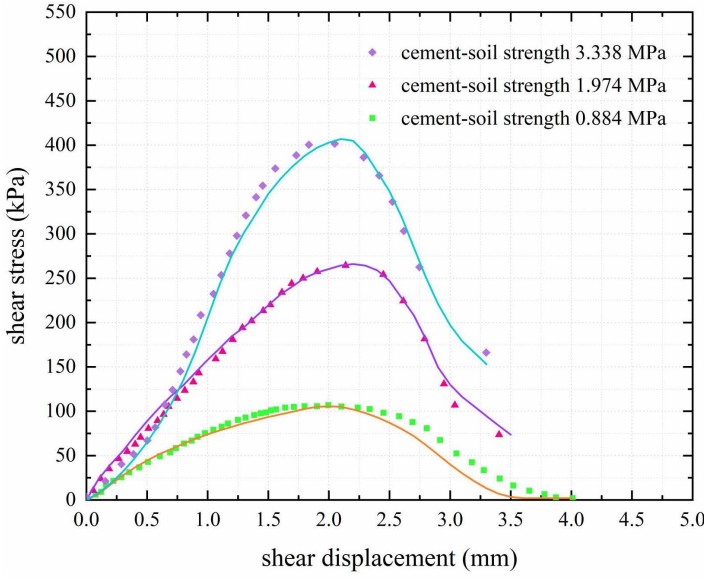

**Fig 8. PC pile–cemented soil interface shear test results.**

The shear stress-displacement curves under different cemented soil strengths are in good agreement with the exponential damage model (scattered points represent experimental data, lines represent fitted model curves), with $R^2$ values of 0.987, 0.990, and 0.998 for the three groups, respectively.

Based on the test results, the average values of peak interfacial shear strength and peak displacement under different conditions were adopted to determine the interfacial ultimate shear strength $\tau_{u1}$ = 250 kPa and the critical displacement $s_{u1}$ = 2 mm. The interfacial stiffness coefficient k1 = 210 MPa/mm was calculated from the pre-peak linear segment, and the Weibull modulus m = 2.0 was obtained from the post-peak softening segment.

**4.2.2 Cemented soil–surrounding soil interface shear test.** By pouring cemented soil together with sand test blocks to form a contact surface, interface shear specimens were prepared. The prepared cemented soil–surrounding soil interface shear specimens are shown in Fig 9. After pouring, all specimens were wet cured for 28 days. Direct shear tests were conducted on the specimens using a ZJ-type strain-controlled direct shear apparatus in accordance with the GB/T 50123−2019 standard. The tests were carried out under normal stresses of 50, 100, 150, and 200 kPa, with a shear displacement rate of 0.8 mm/min. When the shear force was applied, the lower shear box moved to the right, causing relative displacement between the upper and lower shear boxes, thereby resulting in shear failure of the specimen along the contact surface between the cemented soil and the soil.

The results show that, as presented in **Fig 10**, under different cemented soil strengths and normal stresses, the shear stress-displacement curves of the cemented soil–surrounding soil interface conform to the ideal elastoplastic model. The scattered points represent the experimental data, and the lines represent the fitted model curves. The $R^2$ values for the four groups are 0.952, 0.983, 0.990, and 0.978, respectively.

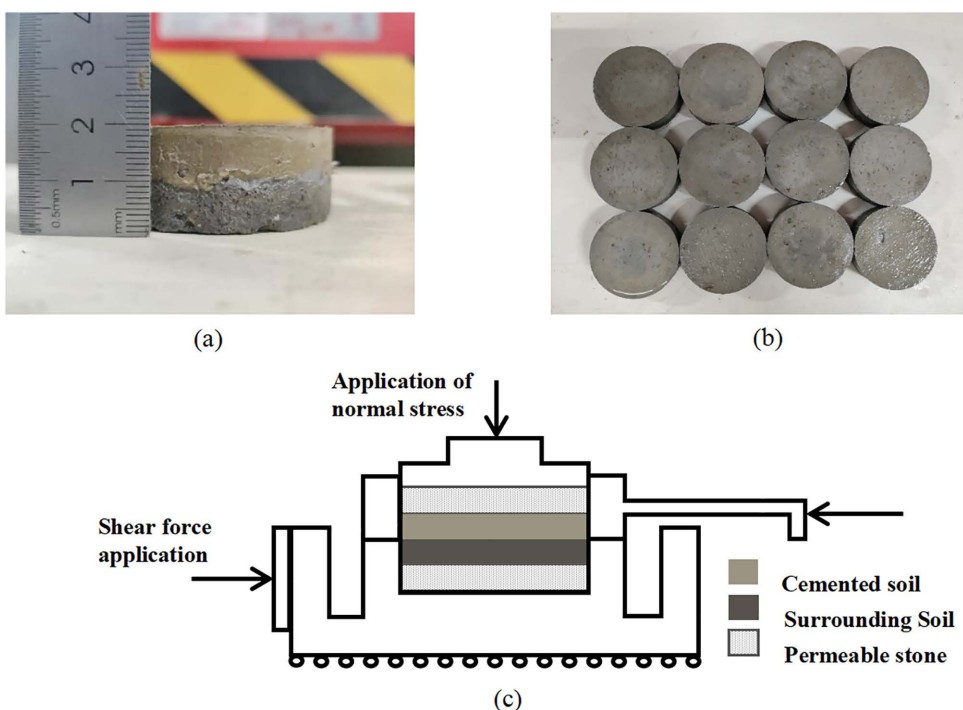

(a)

(b)

(c)

**Fig 9. Specimen and shear device (a) Cemented soil–surrounding soil interface (b) Specimen preparation (c) ZJ-type strain-controlled direct shear apparatus.**

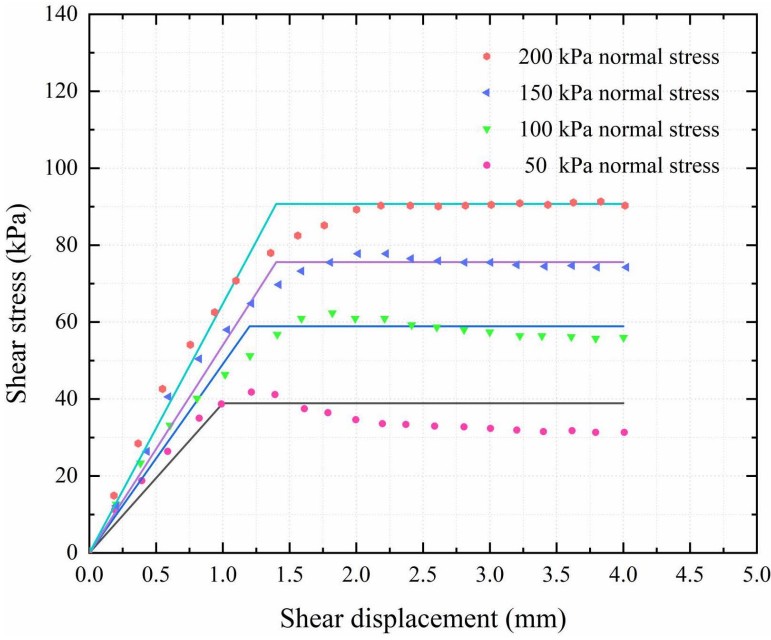

**Fig 10. Cemented soil–surrounding soil interface shear test results.**

The interface parameters in this study are determined based on the sand strata of the engineering site. For clay or silty clay strata, corresponding interface parameters can be adopted in the proposed method. Based on the test results, the average values of peak interfacial shear strength and peak displacement under different working conditions are adopted to determine the parameters. The ultimate skin friction for the sand layer is determined as $\tau_{u2}$ = 65 kPa, with a critical displacement $s_{u2}$ = 1.5 mm.

## 5 Engineering case verification and analysis

### 5.1 Engineering case verification

The accuracy of the proposed theoretical solution was validated against field monitoring data from a PCCS project in Jiangxi Province. Key parameters include: pile length L = 14 m; outer cemented soil diameter $D_c$ = 850 mm; inner PC pile (C105 concrete) diameter $D_p$ = 500 mm; elastic moduli $E_p$ = 44.913 GPa and $E_c$ = 1.458 GPa (determined from laboratory compression tests based on prototype conditions). For the sandy soil, G = 4.614 MPa and $\nu$ = 0.3.

The interface parameters are assigned as follows. For the PC pile–cemented soil interface, based on direct shear tests, the ultimate interfacial shear strength is $\tau_{u1}$ = 250 kPa and the critical relative displacement is $s_{u1}$ = 2 mm. For the cemented soil–surrounding soil interface, based on direct shear tests, the ultimate interfacial shear strength is $\tau_{u2}$ = 65 kPa and the critical relative displacement is $s_{u2}$ = 1.5 mm.

As shown in Fig 11, the theoretical load-settlement (Q–S) curve shows strong agreement with the measured data, as reflected by a coefficient of determination (R²) of 0.9746. Both curves exhibit a gradual increase in settlement up to a load of 6800 kN, followed by a sharp rise at 7650 kN, indicating failure and defining the ultimate bearing capacity. At the maximum applied load of 8500 kN, the relative error in settlement at the maximum load is 13.5%. This discrepancy arises from the simplification of the pile tip as an elastic half-space and the inherent variability of geotechnical materials. Nevertheless, this minor deviation does not undermine the overall validity of the model. The high coefficient of determination across the

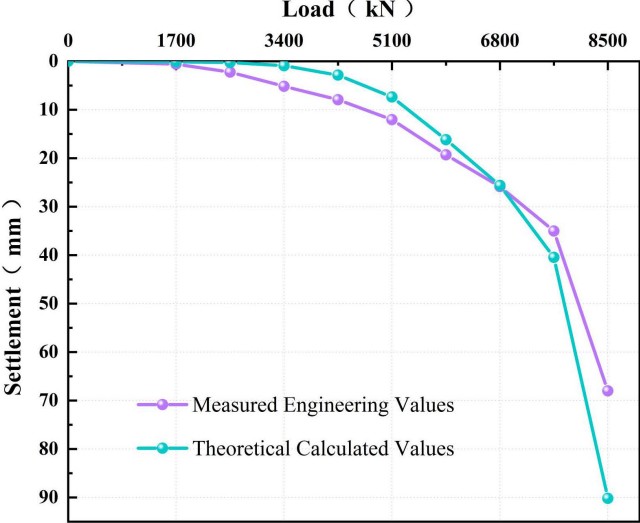

**Fig 11. Comparison of measured and calculated load-settlement (Q-S) curves.**

entire loading range demonstrates that the model captures the coupled dual-interface load-transfer mechanism. Given the inherent variability of geotechnical materials, this accuracy is acceptable for engineering practice, confirming the reliability of the model for analyzing PCCS.

### 5.2 Analysis of load-transfer mechanism

Using the engineering case validated in Section 5.1, theoretical calculations are performed to obtain the axial force distributions in the PC pile and cemented soil, as well as the corresponding skin friction profiles along the depth. This enables an in-depth investigation of the load-transfer mechanism in PCCS foundations.

**5.2.1 Analysis of axial force along pile shaft.** Fig 12 shows the nonlinear decrease in axial force along the PC pile shaft under stepwise loading. The axial force peaks at a depth of approximately 3 m, corresponding to the neutral plane of the pile-soil system. In the shallow depth range of 0–3 m, the cumulative consolidation settlement of the cemented soil exceeds the rebound of the PC pile. This causes reverse shear slip at the PC pile–cemented soil interface, inducing negative skin friction and a cumulative increase in the PC pile axial force. In contrast, in the deeper range of 4–14 m, the penetration settlement of the PC pile dominates the soil displacement, and the gradual mobilization of positive skin friction leads to axial force dissipation.

Furthermore, a significant disparity exists in load distribution at the pile head: the PC pile carries 90.72% of the vertical load, whereas the cemented soil bears only 9.17%. This demonstrates the dominant load-carrying role of the high-stiffness PC pile. With increasing depth, load is progressively transferred from the PC pile to the cemented soil via interfacial friction, causing the load share of the PC pile to decrease gradually [37]. This trend underscores the governing role of interfacial shear stress in mobilizing the bearing capacity of the foundation.

Fig 13 reveals that the cemented soil exhibits a characteristic C-shaped axial force distribution along the pile depth. This C-shaped pattern arises because negative skin friction and interfacial slip in the shallow layer reduce the axial force, while enhanced shear hardening and positive skin friction in the deep layer increase it again. Initially, its load share is less than 11% at the pile head but increases significantly to approximately 40% near the pile toe. This distinctive distribution results from the coupling between material stiffness contrast and interfacial shear

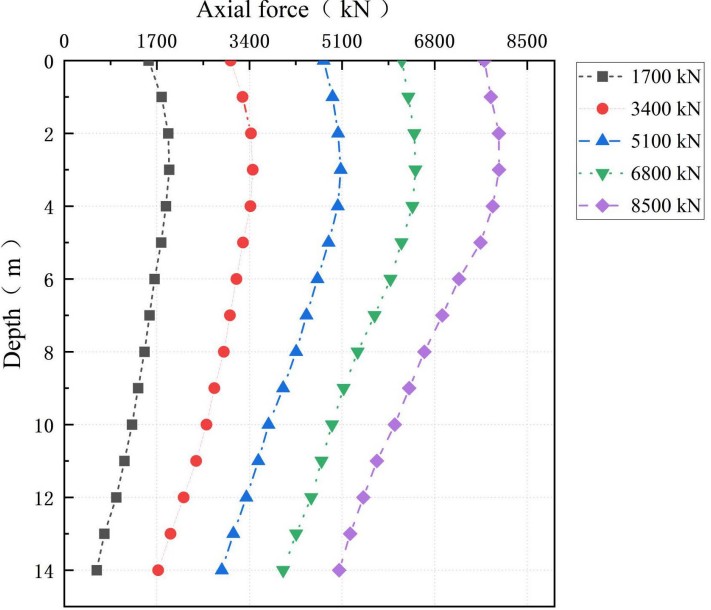

**Fig 12. Axial force distribution along the PC pile shaft.**

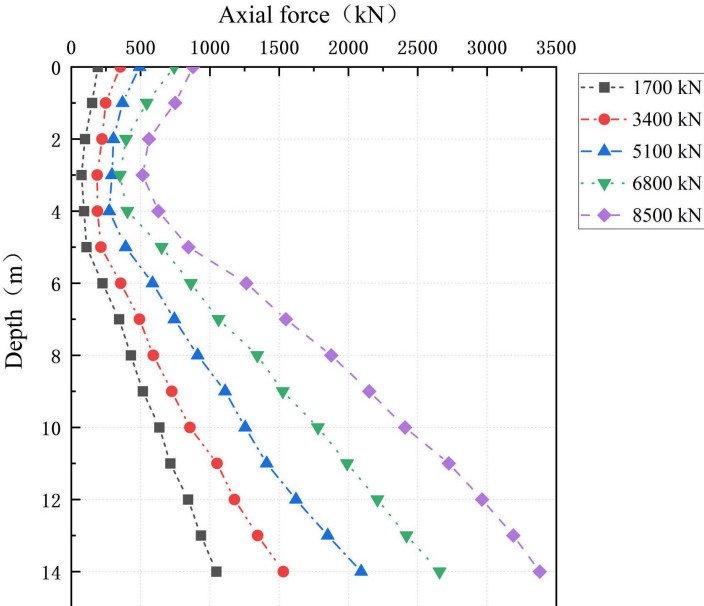

**Fig 13. Axial force distribution in the cemented soil.**

stress. The underlying mechanisms can be delineated by depth. Near the pile head, the cemented soil initially shares load through interfacial bonding. However, within the 0–3 m depth range, load is progressively transferred to the higher-stiffness PC pile due to interfacial slip and stiffness mismatch, consequently reducing the axial force in the cemented soil. Conversely, from 3 to 14 m depth, increasing confining pressure and crack development induce pronounced shear hardening at the cemented soil–surrounding soil interface. This mechanism enhances deep friction mobilization, allowing the cemented soil to carry a higher axial force again and forming the ascending segment of the C-shaped curve.

Moreover, the opposing trends in axial force (decreasing in the PC pile and increasing in the cemented soil) intensify with increasing load. This behavior is driven by soil-structure reorganization under radial compaction: accumulated confining pressure enhances drainage and strengthens interfacial bonding via roughness effects.

**5.2.2 Analysis of skin friction.** The skin friction distribution at the PC pile–cemented soil interface is analyzed using Figs 14 and 15. Governed by the neutral-plane mechanism, negative friction develops in the shallow zone (0–3 m) where cemented soil settlement exceeds PC pile rebound. Conversely, positive friction is mobilized in the deep zone (4–14 m) due to greater PC pile penetration relative to cemented soil creep. At a depth of approximately 3 m, the interface exhibits negligible relative displacement and near-zero friction. Analysis across load levels reveals that the neutral plane position remains stable, independent of loading magnitude, and is determined solely by the relative displacement distribution between the pile and cemented soil.

In the PCCS, the coupled interaction at its dual interfaces governs both the axial force in the cemented soil and the development of skin friction. The neutral plane at the cemented soil–surrounding soil interface varies between 3 and 6 m with load. Above this plane, the surrounding soil shears the cemented soil downward, while the PC pile shears it upward; below this plane, these shear directions are reversed. Moreover, the skin friction at both interfaces exhibit synchronous softening, indicating a coupled progressive failure mechanism inherent to the PCCS system.

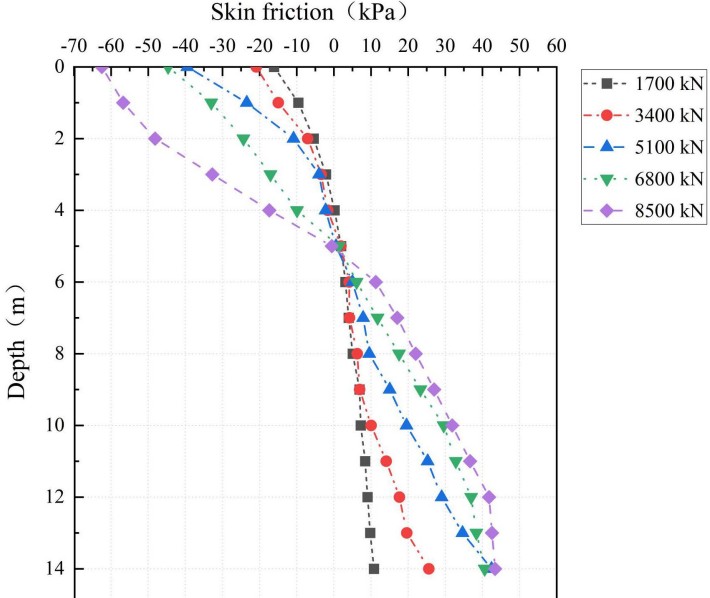

**Fig 14. Skin friction at the PC pile-cemented soil interface.**

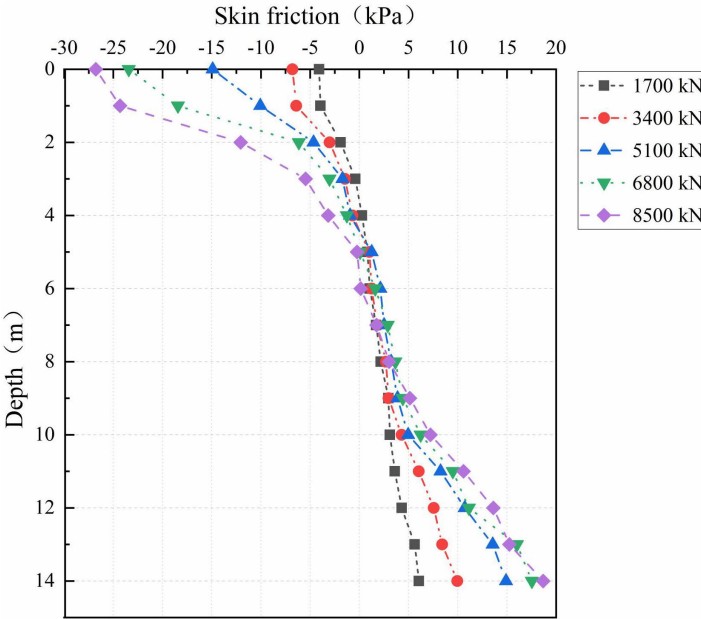

**Fig 15. Skin friction at the cemented soil-surrounding soil interface.**

## 6 Conclusions

This paper presents a dual-interface load transfer model for PCCS that incorporates progressive damage at the PC pile–cemented soil interface. The main findings are summarized as follows:

(1) The exponential damage model captures the nonlinear deformation and stiffness degradation observed at the PC pile–cemented soil interface in direct shear tests. When coupled with an elastoplastic model for the outer interface, the combined formulation realistically represents the dual-interface system in PCCS.

(2) The proposed model achieves high predictive accuracy, with a coefficient of determination ($R^2$) of 0.9746 between the calculated and measured load-settlement curves, demonstrating strong predictive capability.

(3) Under vertical loading, the high-stiffness PC pile carries more than 90% of the applied pile-head load. Load is progressively transferred to the cemented soil and surrounding soil through shear stress mobilization at both interfaces.

(4) A stable neutral plane exists at approximately 3 m depth for the PC pile–cemented soil interface, with negative skin friction above and positive friction below. The synchronous softening of skin friction at both interfaces confirms the coupled progressive failure mechanism inherent to the PCCS system.

(5) The cemented soil exhibits a characteristic C-shaped axial force distribution along the pile, with a low load share at the pile head and a substantially increased share near the toe. This behavior arises from the combined effects of material stiffness contrast and confining pressure-dependent interface response.

The proposed model and algorithm provide a reliable theoretical tool for the detailed design, bearing capacity evaluation, and settlement prediction of PCCS. A limitation of this study is that the validation was conducted using only a single test pile under specific working conditions. The primary contribution of this paper is to propose a theoretical computational algorithm, and this limitation does not compromise the validity of the proposed theoretical framework. Future work will address this limitation through validation under multiple working conditions.

## Nomenclature

| | |
|---|---|
| $\tau_p$ | Shear stress at PC pile–cemented soil interface |
| $\tau_{u1}$ | Ultimate shear stress at PC pile–cemented soil interface |
| $\tau_r$ | Residual shear strength |
| $k_1$ | Inner interface stiffness coefficient |
| $s_{u1}$ | Critical displacement of inner interface |
| $m$ | Weibull modulus |
| $\tau_c$ | Shear stress at cemented soil–surrounding soil interface |
| $\tau_{u2}$ | Ultimate shear stress at cemented soil–surrounding soil interface |
| $s_{u2}$ | Critical displacement of outer interface |
| $k_2$ | Outer interface stiffness coefficient |
| $r_m$ | Radius of influence |
| $E_p$ | Elastic modulus of PC pile |
| $E_c$ | Elastic modulus of cemented soil |
| $D_p$ | Diameter of PC pile |
| $D_c$ | Diameter of cemented soil pile |
| $u_p$ | Perimeter of PC pile |
| $u_c$ | Perimeter of cemented soil pile |
| $A_p$ | Cross-sectional area of PC pile |
| $A_c$ | Cross-sectional area of cemented soil |
| $G_s$ | Shear modulus of soil |
| $k_L$ | Stiffness coefficient of soil at pile toe |
| $\nu$ | Poisson's ratio |

## Supporting information

**S1 File. Interface Shear Test Data.** Raw data from the interface shear tests for the PC pile–cemented soil and cemented soil–surrounding soil interfaces.
(XLSX)

**S2 File. Calculated Data.** Calculated data including load-settlement curves, axial forces, and skin friction distributions of the PCCS pile.
(XLSX)

## Author contributions

**Conceptualization:** Yating Zhu.

**Data curation:** Junpeng Zhang.

**Formal analysis:** Hongyan Wu.

**Investigation:** Lanlan Xu, Bin Xu, Kaiwei Cao.

**Writing – original draft:** Haian Liang, Yating Zhu.

**Writing – review & editing:** Haian Liang.

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
