## [Decision Letter · Decision Letter 0]

13 Apr 2026

PONE-D-26-16041A Dual-Interface Load Transfer Model for Precast Concrete-Cored Cemented Soil Piles Considering Progressive DamagePLOS One

Dear Dr. Yating Zhu,

Thank you for submitting your manuscript to PLOS ONE. After careful consideration, we feel that it has merit but does not fully meet PLOS ONE’s publication criteria as it currently stands. Therefore, we invite you to submit a revised version of the manuscript that addresses the points raised during the review process.

We look forward to receiving your revised manuscript.

Kind regards,

Dajiang Geng

Academic Editor

PLOS One

Journal Requirements:

“This work was supported by the National Natural Science Foundation of China (Grant No. 52168045) and the Key Research and Development Program of Jiangxi Province (Grant No. 20252BCF320006).”

“This work was supported by the National Natural Science Foundation of China (Grant No. 52168045) and the Key Research and Development Program of Jiangxi Province (Grant No. 20252BCF320006).”

Please state what role the funders took in the study. If the funders had no role, please state: “The funders had no role in study design, data collection and analysis, decision to publish, or preparation of the manuscript.”

6. In the online submission form, you indicated that The minimal dataset supporting the findings of this study is included within the manuscript. Full raw data will be made available upon reasonable request during peer review, and will be deposited in a public repository upon acceptance for publication.

7. We note that Figure 1 in your submission contain copyrighted image. All PLOS content is published under the Creative Commons Attribution License (CC BY 4.0), which means that the manuscript, images, and Supporting Information files will be freely available online, and any third party is permitted to access, download, copy, distribute, and use these materials in any way, even commercially, with proper attribution. For more information, see our copyright guidelines: http://journals.plos.org/plosone/s/licenses-and-copyright.

Please upload the completed Content Permission Form or other proof of granted permissions as an “Other” file with your submission.

Reviewers' comments:

Reviewer's Responses to Questions

**Comments to the Author**

1. Is the manuscript technically sound, and do the data support the conclusions?

Reviewer #1: Partly

Reviewer #2: Yes

Reviewer #3: Yes

2. Has the statistical analysis been performed appropriately and rigorously? 

Reviewer #1: N/A

Reviewer #2: Yes

Reviewer #3: Yes

3. Have the authors made all data underlying the findings in their manuscript fully available?

Reviewer #1: Yes

Reviewer #2: Yes

Reviewer #3: Yes

4. Is the manuscript presented in an intelligible fashion and written in standard English?

Reviewer #1: Yes

Reviewer #2: Yes

Reviewer #3: Yes

5. Review Comments to the Author

Reviewer #1: Reviewer Comments

The paper entitled A Dual-Interface Load Transfer Model for Precast Concrete-Cored Cemented Soil Piles Considering Progressive Damage presents a dual-interface load-transfer model that accounts for progressive damage at the precast concrete (PC) core–cemented soil interface. Combined with laboratory test calibrations and field engineering validation, the research topic is highly relevant to studies on the load-transfer mechanism of composite pile foundations in geotechnical engineering, and offers potential value for the engineering design and bearing capacity evaluation of precast concrete-cored cemented soil (PCCS) piles. Nevertheless, several points in the manuscript require further clarification and improvement, as detailed below:

The introduction lacks sufficient clarity in identifying the specific research gaps. The manuscript should clearly state the key differences between the proposed “coupled exponential damage model and elastoplastic model” and existing PCCS load-transfer models in the literature, and discuss relevant pioneering work accordingly. It should also explicitly point out the limitations of previous models in characterizing interface damage and dual-interface coupling mechanisms, as well as the specific approaches adopted in this study to overcome these limitations.

The current reference list is dominated by citations from Chinese scholars. A more diverse range of international references, including those from international academic conferences, should be incorporated to reflect the latest global progress in PCCS, composite piles, and interface damage modeling, thereby enhancing the international perspective and academic relevance of the study.

The decomposition expression for pile-head settlement in Equation (1) (“S” (“z” )“ =” 〖〖“ s” 〗_“pc” “ + s” 〗_“cs” “ + ” “S” _“s” ) has been adopted in several previous studies. It is suggested that a proper citation be added to clarify the origin of this expression and strengthen the rigor of the theoretical derivation.

The manuscript employs a variety of symbols that are critical to the model description. A comprehensive table of symbols and parameters is recommended, listing the name, physical meaning, and unit of each symbol and parameter to improve readability.

The specimen details in the interface shear tests are relatively brief. Additional experimental information, such as the curing conditions of specimens, may be supplemented.

The mechanism analysis of axial force along the pile shaft and shaft friction lacks sufficient depth. The manuscript only reports the observed “C-shaped distribution” of axial force along depth but does not explain the underlying causes, such as the stiffness difference between the PC core and cemented soil, the non-uniform mobilization of shear stress at the two interfaces, and the variation in the proportion of pile-tip resistance. A brief mechanistic explanation is suggested.

Reviewer #2: This manuscript proposes a dual-interface load-transfer framework that considers progressive damage at the inner interface, and the engineering motivation is clear. The manuscript can be considered for publication after minor revision, provided that the authors address the following points to improve clarity and consistency of interpretation.

1. Lines 69-70: Some claims are too strong. It states that the iterative algorithm has “guaranteed convergence.” The “guaranteed convergence” should be revised unless a formal proof or additional convergence study is provided.

2. The assumptions, considering the pile, cemented soil and surrounding soil as homogeneous, isotropic, and linearly elastic materials, are acceptable for a simplified analytical model. However, the authors should briefly discuss the limitation of this assumption and clarify the scope of applicability more explicitly.

3. Line 249 and Line 271: The determination of the critical relative displacements s_u1and s_u2could be clarified further. Since the interface test curves may vary slightly under different test conditions, the authors are encouraged to explain how these representative values were determined. Please clarify whether they correspond to peak-displacement values, fitted transition points, or averaged values from multiple tests.

4. The Introduction could be improved by discussing some closely related studies. I suggest the authors cite and discuss the following relevant paper:

Li, L., Lai, N., Zhao, X., Zhu, T., & Su, Z. (2023). A generalized elastoplastic load-transfer model for axially loaded piles in clay: Incorporation of modulus degradation and skin friction softening. Computers and Geotechnics, 161, 105594.

Ni, P., Song, L., Mei, G., & Zhao, Y. (2017). Generalized nonlinear softening load-transfer model for axially loaded piles. International Journal of Geomechanics, 17(8), 04017019.

5.Line 92: For clarity, please add explicit definitions of all symbols in Eq. (1), including S(z), s_pc, s_cs, and S_s.

6. Lines 152-153: The node numbering scheme is somewhat unconventional, since Node 1 is defined at the pile toe and Node m+1 at the pile head. Please add a brief explanatory note in the text or figure so that readers can follow the recursion procedure more easily

7. Lines 265–266: There is an obvious editing error: “Error! Reference source not found.” This should be corrected.

Reviewer #3: This paper presents a coupled model for the dual-interface load transfer of Precast Concrete-Cored Cemented Soil Piles (PCCS), which considers progressive damage at the inner interface and elastoplastic behavior at the outer interface. The model parameters are calibrated via direct shear tests and validated using field pile data, with a coefficient of determination R2=0.9746. The overall work is solid and practically valuable for engineering applications. However, several issues require clarification and revision by the authors:

1. The PC pile, cemented soil, and surrounding soil are assumed to be linearly elastic. In reality, plastic deformation and damage occur in cemented soil and surrounding soil under loading. The authors are suggested to explain the rationale for this modeling assumption and indicate whether relevant analyses have been conducted.

2. The specific value of the convergence criterion used in the iterative algorithm is not provided. Additionally, the authors should explain whether the algorithm remains convergent for layered soil conditions.

3. Only the final values of the exponential damage model parameters for the inner interface ( ) are reported. The fitting curves and residuals are not presented, making it impossible to assess the reliability of the calibration.

4. The laboratory direct shear tests were performed on scaled models, which introduce scale effects on material strength and interface properties. The authors should clarify how the scaled parameters can be directly applied to the 14 m full-scale pile.

5. The elastoplastic parameters for the outer interface ( ) are derived solely from sand. The authors should address whether these parameters are applicable to clay or silty clay.

6. The origin of the cemented soil elastic modulus is unclear. The authors must specify whether it was obtained from laboratory tests, empirical values, or literature references.

7. Only one field test pile was used in this study. The small sample size prevents validation of the model’s generalizability to different pile lengths, pile diameters, and soil conditions, and cannot eliminate random errors.

8. The authors are requested to carefully proofread the paper for linguistic polishing and correct all formatting errors, including incorrect citations (e.g., “Error! Reference source not found”).

6. PLOS authors have the option to publish the peer review history of their article (what does this mean?). If published, this will include your full peer review and any attached files.

Reviewer #1: No

Reviewer #2: No

Reviewer #3: No

---

## [Author Response · Author response to Decision Letter 1]

21 Apr 2026

Dear Editor and Reviewers,

Thank you for your decision letter regarding our manuscript (PONE-D-26-16041). We sincerely thank the editorial personnel and the reviewers for their professional review work, constructive comments, and valuable suggestions on our article.

The following changes have been made in response to the journal requirements:

1. The copyright issue of Figure 1 has been resolved (replaced with an original schematic diagram).

2. The funding statement and funder role have been amended in the cover letter.

3. The data availability statement has been updated.

4. The permit information has been added to the Methods section (Section 4.1), stating that no specific permits were required for this study.

5. In response to Reviewer #3's comment, Figs 8 and 10 have been revised. The updated figures have been uploaded.

We sincerely thank the reviewers for their valuable comments, which have helped make our manuscript more rigorous. In the following section, we provide our point-by-point responses to the reviewers' comments.

Sincerely,

Yating Zhu and co-authors

Reviewer #1:

Comment 1: The introduction lacks sufficient clarity in identifying the specific research gaps. The manuscript should clearly state the key differences between the proposed “coupled exponential damage model and elastoplastic model” and existing PCCS load-transfer models in the literature, and discuss relevant pioneering work accordingly. It should also explicitly point out the limitations of previous models in characterizing interface damage and dual-interface coupling mechanisms, as well as the specific approaches adopted in this study to overcome these limitations.

Response 1: Thank you very much for your valuable suggestion. We have revised the introduction to further clarify the research gap, with a focus on elaborating the limitations of existing load transfer models for precast concrete-cored cemented soil (PCCS) piles: most existing models adopt a single-interface assumption, which fails to capture the realistic interaction among the precast concrete core, cemented soil, and surrounding soil; moreover, the interface constitutive models are overly simplified and cannot accurately describe the progressive damage and nonlinear softening characteristics of the dual interfaces under loading. On this basis, we further elucidate the key differences between our proposed dual-interface coupled model and existing models, and explicitly state that by introducing a coupled exponential damage model and elastoplastic model, this study overcomes the aforementioned limitations, thereby making the research motivation and innovations clearer.

Comment 2: The current reference list is dominated by citations from Chinese scholars. A more diverse range of international references, including those from international academic conferences, should be incorporated to reflect the latest global progress in PCCS, composite piles, and interface damage modeling, thereby enhancing the international perspective and academic relevance of the study.

Response 2: Thank you very much for your suggestion. From the perspectives of the dual-interface coupling mechanism of PCCS piles and interface nonlinear softening, we have added two international references [22] and [23] in the introduction. These two references propose a generalized nonlinear softening load transfer model for axially loaded piles and a generalized elastoplastic model considering modulus degradation and skin friction softening, respectively. By incorporating the above references, we have enriched the research perspective on interface damage and load transfer theory in this paper, enhanced the international visibility of the study, and improved the imbalance between domestic and international references in the original manuscript.

Comment 3: The decomposition expression for pile-head settlement in Equation (1) Sz = spc + scs + Ss has been adopted in several previous studies. It is suggested that a proper citation be added to clarify the origin of this expression and strengthen the rigor of the theoretical derivation.

Response 3: Thank you very much for your suggestion. We have added a citation [31] to Equation (1) in Section 2.1.

Comment 4: The manuscript employs a variety of symbols that are critical to the model description. A comprehensive table of symbols and parameters is recommended, listing the name, physical meaning, and unit of each symbol and parameter to improve readability.

Response 4: Thank you very much for your suggestion. We have added a table of Nomenclature, which is placed after the Conclusions section and before the References section.

Comment 5: The specimen details in the interface shear tests are relatively brief. Additional experimental information, such as the curing conditions of specimens, may be supplemented.

Response 5: Thank you very much for your suggestion. We have supplemented the curing conditions in Section 4.2. In the interface shear test section for the precast concrete core–cemented soil interface, we have added the moisture curing condition. In the interface shear test section for the cemented soil–surrounding soil interface, we have also added the moisture curing condition for the specimens. These additions make the experimental details more complete and rigorous.

Comment 6: The mechanism analysis of axial force along the pile shaft and shaft friction lacks sufficient depth. The manuscript only reports the observed “C-shaped distribution” of axial force along depth but does not explain the underlying causes, such as the stiffness difference between the PC core and cemented soil, the non-uniform mobilization of shear stress at the two interfaces, and the variation in the proportion of pile-tip resistance. A brief mechanistic explanation is suggested.

Response 6: Thank you very much for your suggestion. We have deepened the mechanistic analysis of the axial force and shaft resistance along the pile, and have supplemented the explanation of the formation mechanism of the C-shaped distribution in the text (Section 5.2.1). The C-shaped distribution of axial force in the cemented soil arises from depth-dependent variations in load transfer: interface slip and negative skin friction in the shallow section reduce the axial force of the cemented soil, while increased confining pressure and shear hardening at greater depths mobilize positive skin friction, thereby enhancing its bearing capacity.

Reviewer #2:

Comment 1: Lines 69-70: Some claims are too strong. It states that the iterative algorithm has “guaranteed convergence.” The “guaranteed convergence” should be revised unless a formal proof or additional convergence study is provided.

Response 1: Thank you very much for your suggestion. Regarding the overly absolute statement, we have revised the original sentence in the introduction from “An efficient iterative solution algorithm with guaranteed convergence is developed to compute the mechanical response under axial loading” to a more rigorous expression: “An iterative calculation method with convergence characteristics is established to solve the mechanical response under axial loading.”

Comment 2: The assumptions, considering the pile, cemented soil and surrounding soil as homogeneous, isotropic, and linearly elastic materials, are acceptable for a simplified analytical model. However, the authors should briefly discuss the limitation of this assumption and clarify the scope of applicability more explicitly.

Response 2: Thank you very much for your suggestion. We have supplemented the discussion on the limitations of the simplified analytical model assumptions after the basic assumptions section (Section 2.2). Treating the PC pile, cemented soil, and surrounding soil as linearly elastic materials is a common simplification in analytical models of composite piles (see references [28] and [29]). This idealization facilitates theoretical derivation and improves computational efficiency, but it neglects the heterogeneity, anisotropy, and nonlinear characteristics of natural soils. Therefore, the proposed model is suitable for preliminary design and engineering estimation under small deformation conditions.

Comment 3: Line 249 and Line 271: The determination of the critical relative displacements Su1 and Su2 could be clarified further. Since the interface test curves may vary slightly under different test conditions, the authors are encouraged to explain how these representative values were determined. Please clarify whether they correspond to peak-displacement values, fitted transition points, or averaged values from multiple tests.

Response 3: Thank you very much for your suggestion. We have supplemented the explanation in the text (Section 4.2) regarding the determination of Su1 and Su2, which are taken as the peak values of the interface shear curves. Multiple sets of interface direct shear tests under different working conditions were conducted according to actual engineering conditions. The skin friction and corresponding displacement at the peak point for each working condition were extracted, and their average values were taken as the representative values, from which Su1, Su2, τu1 and τu2 were ultimately determined.

Comment 4: The Introduction could be improved by discussing some closely related studies. I suggest the authors cite and discuss the following relevant paper:

Li, L., Lai, N., Zhao, X., Zhu, T., & Su, Z. (2023). A generalized elastoplastic load-transfer model for axially loaded piles in clay: Incorporation of modulus degradation and skin friction softening. Computers and Geotechnics, 161, 105594.

Ni, P., Song, L., Mei, G., & Zhao, Y. (2017). Generalized nonlinear softening load-transfer model for axially loaded piles. International Journal of Geomechanics, 17(8), 04017019.

Response 4: Thank you very much for your suggestion. We have cited and discussed the two papers recommended by the reviewer as references [22] and [23] in the introduction. These two references propose load transfer models considering nonlinear softening and modulus degradation for axially loaded piles, respectively, which enrich the existing research on load transfer models incorporating interface softening characteristics. The relevant findings have been reviewed and summarized in the research background of this paper.

Comment 5: Line 92: For clarity, please add explicit definitions of all symbols in Eq. (1), including Sz , spc, scs, and Ss.

Response 5: Thank you very much for your suggestion. We have added the definitions of all symbols in Equation (1) below the equation in Section 2.1.

Comment 6: Lines 152-153: The node numbering scheme is somewhat unconventional, since Node 1 is defined at the pile toe and Node m+1 at the pile head. Please add a brief explanatory note in the text or figure so that readers can follow the recursion procedure more easily.

Response 6: Thank you very much for your suggestion. We have supplemented the explanation of the recursive calculation procedure in Section 3.2. The pile is divided into m equal-length elements, generating a total of m+1 nodes. Among these, Node 1 is located at the pile toe and Node m+1 at the pile head. The recursive analysis proceeds upward from the pile toe, because the displacement at the pile toe is generally the smallest along the entire pile shaft. In this model, the analysis is initiated with a specified pile toe displacement as the boundary condition. Performing the recursive calculation upward from the pile toe minimizes the error introduced by the assumed displacement boundary condition, thereby more accurately reflecting the actual mechanical response of the pile.

Comment 7: Lines 265–266: There is an obvious editing error: “Error! Reference source not found.” This should be corrected.

Response 7: Thank you very much for your suggestion. We have corrected the citation error that appeared in the manuscript and carefully proofread all citations throughout the paper.

Reviewer #3:

Comment 1: The PC pile, cemented soil, and surrounding soil are assumed to be linearly elastic. In reality, plastic deformation and damage occur in cemented soil and surrounding soil under loading. The authors are suggested to explain the rationale for this modeling assumption and indicate whether relevant analyses have been conducted.

Response 1: Thank you very much for your suggestion. We have supplemented the analysis and discussion of the limitations of the relevant simplifying assumptions in the “Basic Assumptions” section (Section 2.2). Simplifying the PC pile, cemented soil, and surrounding soil as linearly elastic materials highlights the nonlinear mechanical behavior of the interfaces and ensures computational efficiency and engineering practicality of the model. This simplification is also a common approach in analytical models of composite piles (references [28] and [29]). Although this idealization facilitates theoretical derivation, it neglects the heterogeneity, anisotropy, and nonlinear characteristics of natural soils. Therefore, the model is more suitable for preliminary design and engineering estimation under small deformation conditions.

Comment 2: The specific value of the convergence criterion used in the iterative algorithm is not provided. Additionally, the authors should explain whether the algorithm remains convergent for layered soil conditions.

Response 2: Thank you very much for your suggestion. We have supplemented the specific value of the convergence criterion in the text (Section 3.3) and explained that the algorithm remains convergent under layered soil conditions. The iterative calculation in this paper adopts a relative error convergence criterion, with the convergence tolerance controlled within 1% to 5%, which meets the general requirements for engineering analysis and design. The algorithm is also applicable to layered soils; it only requires updating the corresponding soil parameters according to the layer thickness and continuing the iteration. Since the engineering case in this paper primarily involves a sandy soil foundation, the calculation example uses a single-layer soil for analysis.

Comment 3: Only the final values of the exponential damage model parameters for the inner interface ( ) are reported. The fitting curves and residuals are not presented, making it impossible to assess the reliability of the calibration.

Response 3: Thank you very much for your suggestion. We have supplemented the fitting curves (Figs 8 and 10) in the revised manuscript, and added the residual analysis results in Sections 4.2.1 and 4.2.2. To evaluate the reliability of the parameter calibration, we provide the residual analysis results below. For the PC pile–cemented soil interface (exponential damage model), the R² values for the three datasets are 0.987, 0.990, and 0.998, with RMSE values of 7.32, 17.57, and 10.07 kPa, and mean residual values of +3.68, -13.75, and +4.46 kPa, respectively. For the cemented soil–surrounding soil interface (elastic-perfectly plastic model), the R² values for the four datasets are 0.952, 0.983, 0.990, and 0.978, with RMSE values of 4.60, 2.93, 3.10, and 4.24 kPa, and mean residual values of -2.05, -0.46, +0.79, and -0.58 kPa, respectively. The above results are the supplementary outcomes of the residual analysis.

Comment 4: The laboratory direct shear tests were performed on scaled models, which introduce scale effects on material strength and interface properties. The authors should clarify how the scaled parameters can be directly applied to the 14 m full-scale pile.

Response 4: Thank you very much for your suggestion. In this study, the 14 m full-scale pile was scaled down according to the similarity theory, with a geometric similarity ratio of 1:14 and an elastic modulus similarity constant of 1.446. Both geometric and physical-mechanical similarity relationships were satisfied. Dimensionless parameters such as strain and Poisson's ratio can be directly applied to the analysis of the full-scale pile. The peak shear stress and peak displacement at the interface are primarily controlled by the material mix proportion, interface roughness, and normal stress, and the scaled tests were consistent with the prototype engineering conditions in these respects. Due to limitations in test conditions,

---

## [Decision Letter · Decision Letter 1]

6 May 2026

A Dual-Interface Load Transfer Model for Precast Concrete-Cored Cemented Soil Piles Considering Progressive Damage

PONE-D-26-16041R1

Dear Dr. Zhu,

We’re pleased to inform you that your manuscript has been judged scientifically suitable for publication and will be formally accepted for publication once it meets all outstanding technical requirements.

An invoice will be generated when your article is formally accepted. Please note, if your institution has a publishing partnership with PLOS and your article meets the relevant criteria, all or part of your publication costs will be covered. Please make sure your user information is up-to-date by logging into Editorial Manager at Editorial Manager® and clicking the ‘Update My Information’ link at the top of the page. For questions related to billing, please contact billing support.

Kind regards,

Dajiang Geng

Academic Editor

PLOS One

Additional Editor Comments (optional):

Reviewers' comments:

Reviewer's Responses to Questions

**Comments to the Author**

1. If the authors have adequately addressed your comments raised in a previous round of review and you feel that this manuscript is now acceptable for publication, you may indicate that here to bypass the “Comments to the Author” section, enter your conflict of interest statement in the “Confidential to Editor” section, and submit your “Accept” recommendation.

Reviewer #1: All comments have been addressed

Reviewer #2: All comments have been addressed

2. Is the manuscript technically sound, and do the data support the conclusions?

Reviewer #1: Yes

Reviewer #2: Yes

3. Has the statistical analysis been performed appropriately and rigorously? 

Reviewer #1: Yes

Reviewer #2: N/A

4. Have the authors made all data underlying the findings in their manuscript fully available?

Reviewer #1: Yes

Reviewer #2: Yes

5. Is the manuscript presented in an intelligible fashion and written in standard English?

Reviewer #1: Yes

Reviewer #2: Yes

6. Review Comments to the Author

Reviewer #1: The authors have revised their manuscript as requested, and it has significantly improved compared to the original version. Therefore, I recommend accepting and publishing it.

Reviewer #2: The authors have adequately addressed my previous comments. The revised manuscript is now acceptable for publication. I have no further concerns and recommend acceptance.

7. PLOS authors have the option to publish the peer review history of their article (what does this mean?). If published, this will include your full peer review and any attached files.

Reviewer #1: No

Reviewer #2: No
